# Effects of Organizational Leadership on Project Citizenship Behavior and Management Performance in Complex Construction Projects

**Lan Luo [1], Yue Yang [2], Guangdong Wu [3],\*, Junwei Zheng [4] and Defa Liu [2]**

[1] School of Public Policy and Administration, Nanchang University, Nanchang 330031, China
[2] School of Infrastructure Engineering, Nanchang University, Nanchang 330031, China
[3] School of Public Policy and Administration, Chongqing University, Chongqing 400044, China
[4] Kunming University of Science and Technology, Kunming 650093, China
\* Correspondence: gd198410@163.com

**Abstract:** Organizational leadership is a key factor affecting the management performance of complex construction projects, but seldom have studies attempted to explore the effect mechanisms of organizational leadership on the project management performance, especially the mediating role of project citizenship behavior. The purpose of this study is to fill this gap by investigating the effects of organizational leadership on project citizenship behavior and management performance in complex construction projects. The theoretical model is constructed based on a literature review, and exploratory factor analyses (EFA) are performed on 169 valid questionnaires collected to measure organizational leadership, then partial least squares-structural equation modeling (PLS-SEM) is used to test the hypotheses. The results show that (i) organizational leadership is measured as vision guiding, context interacting, team building, and systems thinking; (ii) vision guiding and context interacting have both direct and indirect effects on the project management performance, and team building can only improve the project management performance by influencing the project citizenship behavior, whereas systems thinking has no significant effect on project citizenship behavior and the project management performance in complex construction projects; (iii) and project citizenship behavior partly mediates the influence of organizational leadership on the project management performance, and the effect of organizational leadership on the project management performance is more realized through the mediating role of project citizenship behavior. The results have a significant theoretical and practical significance for improving the project management performance.

**Keywords:** complex construction projects; organizational leadership; project management performance; project citizenship behavior





## 1. Introduction

With the acceleration of urbanization and the rapid development of the scientific and technological innovation ability, complex construction projects increase rapidly all over the world. Complex construction projects are characterized by a large scale, a large number of elements, interactions, uncertainty, and dynamics. The high level of project complexity poses a key challenge to the successful execution of the project [1]. For example, the Wudongde hydropower plant with a total investment of CNY 73.813 billion and a construction period of 114 months, has a large number of crossover operations and many parties are involved in the construction. The more complex the project the project members face is, the less prior experience they can use [1,2]. Due to the complexity of complex construction projects, the difficulty of construction, and the ambiguity of a risk perception of all parties involved, it provides significant obstacles for project managers, resulting in catastrophic implications such as investment overruns, timetable delays, and other out-of-control ambitions [3–5]. Moreover, COVID-19 adds urgency and temporality to

the management of complex construction projects. Within the context of the COVID-19 unprecedented crisis, project managers have to face new challenges and adapt to a work environment with fewer social interactions [6].

Leadership is an essential aspect influencing the success of construction projects, and research supports the link between leadership and project performance [7,8]. During the epidemic, leaders have had to confront unexpected changes in the social and economic crisis, which requires strong organizational leadership. The temporary nature of the arrangements made during COVID-19 (such as project suspension, the readjustment of workers to new social distances, the introduction of new protective measures, etc.) places greater demands on organizational leadership.

In addition, leadership has been viewed as a crucial component for organizations, and its guiding effect on employees' behavior is a very interesting topic. Meanwhile, the uncontrollable nature of humans becomes an important influence factor in complex construction projects during the epidemic. Leaders guide their teams through organizational leadership to stimulate project citizenship behaviors to face uncertainty. A review by Yang et al. [8] showed that there is intense contact between project citizenship behavior and the project management performance. For example, the project management performance can be improved by increasing trust between employees and management, enhancing coordination, and improving the ability to adapt to unexpected situations. Shafi et al. [9] analyzed the project success measure assessment of major infrastructure projects based on the theory of project citizenship behavior and confirmed that project citizenship behavior is positively correlated with project success through the method of fuzzy mathematics. Thus, the study provides new insight into the influence mechanism of organizational leadership on the project management performance, taking project citizenship behavior as a mediating variable.

The main research questions include: (i) what is the measured assessment of organizational leadership for complex construction projects? (ii) What are the effects of organizational leadership dimensions on the project management performance? (iii) What is the mediating role of project citizenship behavior between the effects? The purpose of this study is to develop the measurement of organizational leadership for complex construction projects and explore the relationship between organizational leadership, the project management performance, and project citizenship behavior. The study offers substantial theoretical and practical insights into organizational leadership development and complex project management and provides new ideas for improving the management performance of complex construction projects under the impact of COVID-19.

## 2. Theoretical Background

### 2.1. Organizational Leadership (OL)

Organizational leadership is derived from leadership theory in management [10]. Existing research of leadership theory focused on transformational leadership [10–12], visionary leadership [13,14], and organizational leadership [15–17]. Leadership is the ability of a leader to influence and lead followers to achieve a vision. It is a core competency of a leader to use leadership resources to achieve a vision in a certain situation [10]. Stephen and Richard [18], from the perspective of context, defined organizational leadership as the necessary leadership behavior of senior management determined by the problems faced by the organizational environment, and finally achieved the goals of the organization by responding to the demands of the situation. Palaima [19] developed a theoretical model of organizational leadership, including the personal leadership dimension, relational leadership dimension, and organizational/strategic leadership dimension. Organizational leadership was first introduced by Velsor and Mccauley [20] and was defined as the ability of an organization to drive its members to accomplish collective tasks when faced with the challenges of change. Similarly, Day [16] argued that there are three levels of organizational leadership, including the leader level, the follower, peer, and supervisor relationship level, and the organizational culture level. In addition, a measured assessment of organizational

leadership in the enterprise was proposed by Kivipõld et al. [21] encompassing vision orientation, team building, cultural sensitivity, environmental interaction, organizational support, and organizational operation. It is worth noting that a vision is different from a goal that defines specific outcomes [22]. Vision is defined as the mental picture that organizational members have of the desired future of the organization [14]. Empirically, organizations are systems that pursue goals rather than achieve them; that is, they constantly seek and change them [14]. Thus, the function of the vision is crucial.

In sum, organizational leadership is the latest development in leadership theory. Combining the existing research theories, organizational leadership is defined as the sum of individual leadership within an organization and the influence generated by the interaction of the individual, team, and situational factors [18,23]. Meanwhile, it is a multidimensional concept of vision guiding, team building, and organizational support [23]. Organizational leadership combines the advantages of organizational behavior and leadership, which has a particularly significant impact on the management of complex construction projects. Compared to general leadership, complex construction project leadership should be capable of context interaction and systematic thinking [24]. Accordingly, it is necessary to systematically define organizational leadership based on the features of complex construction projects.

### 2.2. Project Citizenship Behavior (PCB)

Project citizenship behavior is derived from the concept of organizational citizenship behavior. According to Organ [25], organizational citizenship behavior was a sum of behaviors that can effectively promote the operation of the organization. Early researchers of organizational citizenship behavior considered citizenship behavior as separate from the in-role job performance. They emphasized that organizational citizenship behavior should be viewed as an extra-role function and an organizational function [25,26]. Extending to the field of construction projects, the concept of project citizenship behavior was proposed by Braun et al. [27] and further deepened into the concept of temporary organizational citizenship behavior. Project citizenship behavior is considered to be a behavior in that project personnel focus on inter-organizational cooperation and adopt effective actions to improve the project performance [27]. He et al. [28] defined project citizenship behavior of complex construction projects as positive and free behavior that is not written in the contracts or specified in a uniform statement by project management organizations and is generally conducive to effectively realizing construction goals. In addition, it includes all positive in-role and extra-role behaviors of individual members of the organization [29].

In conclusion, project citizenship behavior is a voluntary individual behavior that crosses organizational boundaries and is embedded in interpersonal networks [27,30]. It cannot be directly identified by traditional reward systems within projects, however, being effective in enhancing organizational effectiveness [28]. Complex construction projects are difficult to manage. There is an urgent need to transform organizational behavior from a passive task completion to proactive project value realization through self-motivation [28]. Therefore, project citizenship conduct is more crucial to the success of complex construction projects than it is for conventional construction projects.

### 2.3. Project Management Performance (PMP)

Project management performance is an important basis and criterion for project success [31]. Complex construction projects face more severe problems of quality, safety, and cost control, and rely more on scientific decision-making and control implementation in the process of project management. Thereby, it is particularly important to explore the performance improvement path in the construction process to obtain a more effective project management.

Cost, time, and quality are just a few of the key performance indicators (KPIs) used to gauge how well construction projects are performing [31,32]. Almahmoud et al. [33] connected project health to the project performance indicators based on KPIs. Based on the

literature, Chan et al. [34] presented the standard system of a project's success for building projects that took into account the time, cost, quality, health and safety, environmental performance, participant and user satisfaction, and commercial value. It may be stated that over the previous few decades, new KPIs have been proposed to measure a project's success, including the health and safety performance, environmental performance, participants' satisfaction, and client satisfaction [34].

### 2.4. Research Gap in Existing Studies

Existing research [21,35] suggests that there is a strong relationship between organizational leadership and the performance of the management team. However, few studies have attempted to examine the specific ways in which such effects occur based on a complexity perspective. Complex construction projects are more uncertain, more complex, and more difficult to implement than general projects [1,2,5]. The mechanisms by which organizational leadership affects the management performance in such contexts remain unclear.

Additionally, COVID-19 has a significant impact on human activities. The impact of project citizenship behavior on performance should not be neglected as well. Some existing studies have used project citizenship behavior as a mediating variable in theoretical models [9,12,36]. Few studies, meanwhile, have attempted to look into how project citizenship behavior may mediate the relationship between organizational leadership and performance. To close the gap, this study considers organizational leadership as a mediating variable and examines its effect on the relationship between organizational leadership and the project management performance.

### 2.5. Hypotheses Development and Theoretical Model

Due to the large scale and uncertainty of complex construction projects, project managers face great challenges when controlling complex projects. In that case, it leads to serious consequences, such as investment overspending and schedule delay. Meanwhile, COVID-19 has severely disrupted the construction industry with serious consequences, such as restricted construction site staffing levels, delayed project schedules, increased financial pressure on companies, and legal difficulties related to the interpretation of contract provisions like force majeure.

#### 2.5.1. Organizational Leadership and Project Management Performance

Organizational leadership in complex construction projects is mainly reflected in team building, spiritual motivation, culture building, vision guiding, team cohesion, and context interacting. The personality traits of the leader [7] and the leadership style of the manager [8,31] all have a direct impact on the project management performance. Strong leadership in the project organization, such as responding well to outbreaks and making quick and accurate decisions, may help to improve the project performance. Thus, the hypothesis is proposed as follows:

**H0:** *Stronger organizational leadership has a significant positive effect on project management performance.*

#### 2.5.2. Organizational Leadership and Project Citizenship Behavior

Organizational leadership promotes organizational citizenship behavior by emphasizing the value of social interaction between leaders and followers in the form of a spiritual contract [12]. Anantatmula [37] showed that the stronger leadership of organizational leaders can motivate organizational members to create a positive working environment. When in the particular situation of COVID-19, the motivational effect of stronger organizational leadership on project employees cannot be ignored. Thus, the hypothesis is proposed as follows:

**H1:** *Stronger organizational leadership has a significant positive effect on project citizenship behavior.*

### 2.5.3. Project Citizenship Behavior and Project Management Performance

Project citizenship behavior helps to meet the "iron triangle" (time, budget, and quality) and improves the relationship between individual participants after the project is completed and maintains it [27]. In the construction process of complex construction projects, project citizenship behavior is conducive to creating a positive organizational situation among all participants in the project [38]. Therefore, the hypothesis is proposed as follows:

**H2:** *Project citizenship behavior has a significant positive effect on project management performance.*

### 2.5.4. Mediating Effect of Project Citizenship Behavior

It was found that project citizenship behavior is treated as a mediating variable in studies investigating employee performance [39], organization performance [40], and individual creativity [41]. In addition, Shafi et al. [9] discussed the relationship between project organizational culture and the project performance, and the results indicate that project organizational culture significantly affects the project performance, while project citizens, as an intermediary variable in the influence process, affect the project performance. This idea is also desirable in the study of the relationship between organizational leadership on the project management performance. When projects have multiple stakeholders, project citizenship behavior may indeed have a positive effect on the project management performance. The goal of improving the project management performance cannot be achieved without strong organizational leadership and the conscious compliance and collaboration of the entire team. Thus, the hypothesis is proposed as follows:

**H3:** *Project citizenship behavior mediates the effects of organizational leadership on project management performance.*

### 2.5.5. Theoretical Model

The leadership of the project manager is crucial in adapting to environmental changes, making quick decisions, inspiring teams, and providing a stable work environment to enhance the project performance, especially in complex construction projects [37]. Fostering employee collectivism and assisting employees in achieving their goals are key organizational leadership principles, thus stronger leadership makes it easier to demonstrate project citizenship behavior among employees [38]. Moreover, organizational citizenship promotes the development of the company's social capital, enhancing employee productivity and organizational effectiveness [25]. In view of this, project citizenship behavior is selected as a mediating variable to explore the impacts of organizational leadership on performance in complex construction projects. The theoretical model is created to explain the connection among organizational leadership, project citizenship behavior, and the project management performance (see Figure 1).

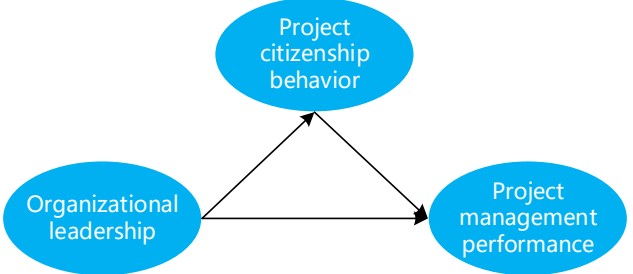

**Figure 1.** Hypothesized theoretical model of OL, PCB, and PMP.

## 3. Method and Data Presentation

### 3.1. Measures and Questionnaire Design

Based on the literature review, the factors of organizational leadership are identified in construction projects with a content analysis. It can be concluded that these factors are mainly analyzed from the perspective of individual characteristics, behaviors, or situations. However, organizational leadership also involves organizational culture, organizational structure, organizational process, and the organizational system at the collective level. Therefore, this study utilized two rounds of Delphi interviews to validate the factors summarized in the literature review so that they would represent the special nature of complex construction projects. The Delphi method is a research technique that entails sending a questionnaire to subject matter experts for feedback. After designing a semi-structured questionnaire and gathering expert input, statistical feedback results are acquired up until the expert opinions are in agreement. The majority of Delphi studies only include up to 20 individuals [42]. Eight professionals with real-world experience are chosen for this poll. The majority of the professionals consulted, including the constructor (1), the owner (3), and the consulting unit (4), have more than ten years of management experience in intricate construction projects (5).

The first round of Delphi conducted an open questionnaire survey on organizational leadership in complex construction projects. Then, the experts were invited to discuss the rationality and accuracy of the measurement indicators in academic seminars. Based on the interview results, the summarized organizational leadership factors are improved. For instance, the interaction is prone to ambiguity and has been modified to the process of roles after discussion. Moreover, two factors have been added to the organizational leadership scale, including cultural identity and information transfer. The organization of complex construction projects is formed by multiple participants on an ad hoc basis, and whether each participant can identify with the organizational culture and follow consistent values is related to organizational leadership. Meanwhile, the internal communications of complex construction projects are multi-dimensional information networks, and the accuracy and timeliness of the information exchange among the participants are related to organizational leadership. In the second round of the Delphi survey, the experts are required to reassess the results in light of the consolidated results obtained in the first round of the survey. The results show that experts reached a consensus on the modified factors of organizational leadership. Thus, a total of 14 potential indicators of organizational leadership are identified and shown in Table 1.

**Table 1.** Organizational leadership measures of complex construction projects from the literature review.

| Factor | Description | References |
|---|---|---|
| Cultural identity (OL1) | All participants in the project have an overall sense of identity in the project culture. | Mueller et al. [7]; Luo et al. [17]; Aarons et al. [23] |
| Development direction (OL2) | All participants in the project have a common direction of development. | Mueller et al. [7]; Luo et al. [17]; Aarons et al. [23]; Ogunlana et al. [43] |
| Personal qualities (OL3) | The members of the project participants have a highly professional and technical level. | Luo et al. [17]; Aarons et al. [23]; Hanna et al. [44] |
| Value orientation (OL4) | The values of all participants in the project are consistent. | Luo et al. [17]; Aarons et al. [23]; Ogunlana et al. [43] |
| Dynamic thinking (OL5) | All participants of the project have made a comprehensive emergency plan to deal with various uncertainties. | Luo et al. [17]; Day et al. [16]; Aarons et al. [23] |
| System integration thought (OL6) | In the face of complex problems, all participants in the project can cooperate to solve them. | Luo et al. [17]; Aarons et al. [23]; Clarke et al. [45] |

**Table 1.** *Cont.*

| Factor | Description | References |
|---|---|---|
| Process of roles (OL7) | All participants in the project are aware of the process of each stage of the project. | Hanna et al. [44] |
| Information transfer (OL8) | The information transmitted between the project participants is timely and fast. | Luo et al. [17]; Aarons et al. [23] |
| Gentle communication (OL9) | The participants of the project can communicate flexibly and create a positive project situation. | Müller et al. [7]; Aarons et al. [23] |
| Resource integration (OL10) | The resources required for each stage of the project can be supplied in time. | Müller et al. [7]; Hanna et al. [44] |
| Strategy adjustment (OL11) | The participants of the project can adjust their strategies flexibly to adapt to the complex environment of the project. | Luo et al. [17]; Aarons et al. [23] |
| Organization structure (OL12) | Your project team has a few levels of organizational structure. | Luo et al. [17]; Hanna et al. [44] |
| Training system (OL13) | The project organization has a perfect training system. | Müller et al. [7] |
| Incentive system (OL14) | The project organization has a good incentive system. | Müller et al. [7]; Luo et al. [17] |

PCB is derived from OCB, according to the studies. As a result, this study reviews relevant publications on the OCB measured assessment and then evaluates the PCB measuring dimensions considering the complexity. Podsakoff et al. [46] proposed the measured assessment of OCB including a helpful conduct, sportsmanship, devotion to one's organization, compliance with the organization, personal initiative, civic morality, and self-improvement. The significant difference between PCB and OCB is that the project organization is temporary and cross-organizational. According to Braun et al. [27], the essence of organizational compliance is to obey organizational rules and regulations and relevant provisions. Therefore, organizational compliance can be summarized as compliance behavior in complex construction projects. The essence of organizational loyalty and sportsmanship is the individual devotion to work, that is, the willingness to work in extreme and unsupervised situations to complete tasks. These two behaviors can be summed up as conscientiousness in complex construction projects. The core of civic morality and politeness is to maintain a harmonious interpersonal relationship. Self-development and individual initiative refer to the ability to creatively complete work or spontaneously improve work skills, which can be explained as innovative behavior in complex construction projects. Helpful conduct refers to offering help to others and the ability of project participants to cooperate to complete tasks. In complex construction projects, He et al. [14] believed that such behavior could be interpreted as collaborative behavior. In sum, the measured assessment of PCB in this study includes compliance behavior (PCB1), conscientiousness (PCB2), harmonious relationship maintenance behavior (PCB3), innovation behavior (PCB4), and collaborative behavior (PCB5).

Regarding the project management performance, the study by Chan et al. [34] is selected as the reference for the questionnaire based on comprehensive existing studies. The evaluation standard system of the project management performance includes time (PMP1), cost (PMP2), quality (PMP3), health and safety (PMP4), environmental performance (PMP5), participants' satisfaction (PMP6), user satisfaction (PMP7), and commercial value (PMP8).

*3.2. Sample and Data Collection*

The information for this study is gathered through a questionnaire survey. Professionals who work as owners, contractors, designers, suppliers, and subcontractors make up

the majority of the survey samples. They are drawn from complex construction projects in China with a construction period of at least 3 months, a cost of at least CNY 1.5 million, a large number of participants, and a high dynamic uncertainty. Non-probability sampling is used in this investigation since a sample frame was not used. Non-probability sampling allows for a representative sample [47]. Survey respondents need to be randomly selected based on their willingness to participate in the study rather than from the entire population.

The questionnaire is created using a five-point Likert scale [48], with 1 denoting strongly disagree and 5 denoting strongly agree, based on the measures. The selection of the respondents is also restricted to ensure the fairness and impartiality of the expert opinions gathered [48]. The specialists are project managers with a minimum of two years of professional experience and are asked to provide information on the most recent challenging construction project that has been completed. The questionnaire star, a reputable online platform for surveys, ratings, and voting in China, has sent out a total of 213 questionnaires. Due to COVID-19, a face-to-face interview is not possible in this study. The interviewees are all willing volunteers, and no gifts or incentives are given to them in order to prevent potential biases. Additionally, the interviewers are allowed flexible time to complete the questionnaire in order to prevent them from using it excessively. The right to revoke participation at any point during the research period has also been made clear to the interviewees. Additionally, interviewees are provided with written informed consent, and confidentiality and anonymity are also guaranteed, which aids in lowering the variation in the common methods.

In this study, SPSS 17.0 and Smart PLS 3.0 software are adopted for analysis, and a significant amount of data must be gathered. Two hundred and thirteen questionnaires were distributed in total and 176 of them were returned. Only 169 of the 176 completed questionnaires were deemed valid since the other 7 are either a duplicate of previous surveys or have missing or incorrect information. The details of the respondents are shown in Table 2 and a summary of the measures is absolute in Table 3. Table 2 reveals that the interviewees are mostly male (84.6%) with bachelor's degrees (45.6%) and more than 10 years of experience working on complex construction projects (31.4%). The majority of the data are gathered from construction projects with budgets greater than CNY 100 million (64.5%) and durations longer than 37 months (46.2 percent). In addition, Independent Sample T Test analysis or a One-Way Analysis of Variance is conducted on respondent characteristics and item characteristics [49]. The results show significant differences in the perception of the project management performance by work experience, project type, and project size ($p < 0.05$). It can be found from Table 3 that the absolute values of the measured values are all less than 3 for skewness and less than 10 for kurtosis. These results demonstrate the sample data's normal distribution compliance and can be used in a further investigation.

**Table 2.** Statistical analysis of the returned 169 valid questionnaires.

| Characteristics | Category | Frequency | Percentage (%) |
|---|---|---|---|
| Education | Ph.D. | 11 | 6.5 |
| | Master's degree | 52 | 30.8 |
| | Bachelor's degree | 77 | 45.6 |
| | Others | 29 | 17.1 |
| Gender | Male | 143 | 84.6 |
| | Female | 26 | 15.4 |
| Work experience | ≤10 years | 116 | 68.6 |
| | 11–15 years | 25 | 14.8 |
| | 16–20 years | 19 | 11.3 |
| | ≥20 years | 9 | 5.3 |

**Table 2.** *Cont.*

| Characteristics | Category | Frequency | Percentage (%) |
|---|---|---|---|
| Project type | Public projects | 48 | 28.4 |
| | Industrial projects | 16 | 9.5 |
| | Others | 105 | 62.1 |
| Project size | ≤50 million CNY | 29 | 17.2 |
| | 50–100 million CNY | 31 | 18.3 |
| | 100–500 million CNY | 61 | 36.1 |
| | >500 million CNY | 48 | 28.4 |
| Project duration | ≤24 months | 50 | 29.5 |
| | 25–36 months | 41 | 24.3 |
| | 37–48 months | 51 | 30.2 |
| | >48 months | 27 | 16.0 |

**Table 3.** Descriptive outline of the measures of OL, PCB, and PMP.

| Measures | Mean | SD | Skewness | | Kurtosis | |
|---|---|---|---|---|---|---|
| | Statistic | Statistic | Statistic | SD | Statistic | SD |
| OL1 | 3.207 | 1.308 | −0.101 | 0.187 | −0.938 | 0.371 |
| OL2 | 3.296 | 1.091 | −0.048 | 0.187 | −0.735 | 0.371 |
| OL3 | 3.414 | 1.316 | −0.17 | 0.187 | −0.901 | 0.371 |
| OL4 | 3.391 | 1.192 | −0.41 | 0.187 | −0.356 | 0.371 |
| OL5 | 3.343 | 1.227 | −0.182 | 0.187 | −0.733 | 0.371 |
| OL6 | 3.645 | 1.230 | −0.399 | 0.187 | −0.704 | 0.371 |
| OL7 | 3.562 | 1.331 | −0.212 | 0.187 | −1.154 | 0.371 |
| OL8 | 3.272 | 1.247 | −0.037 | 0.187 | −0.891 | 0.371 |
| OL9 | 3.485 | 1.287 | −0.223 | 0.187 | −0.961 | 0.371 |
| OL10 | 3.302 | 1.307 | −0.276 | 0.187 | −0.786 | 0.371 |
| OL11 | 3.468 | 1.369 | −0.372 | 0.187 | −0.731 | 0.371 |
| OL12 | 3.331 | 1.366 | −0.152 | 0.187 | −0.937 | 0.371 |
| OL13 | 3.669 | 1.330 | −0.478 | 0.187 | −0.825 | 0.371 |
| OL14 | 3.178 | 1.409 | 0.018 | 0.187 | −1.073 | 0.371 |
| PCB1 | 3.503 | 1.203 | 0.013 | 0.187 | −0.872 | 0.371 |
| PCB2 | 3.633 | 0.813 | −0.209 | 0.187 | −0.209 | 0.371 |
| PCB3 | 3.533 | 0.890 | −0.112 | 0.187 | −0.281 | 0.371 |
| PCB4 | 3.562 | 1.081 | −0.171 | 0.187 | −0.352 | 0.371 |
| PCB5 | 3.574 | 1.221 | −0.270 | 0.187 | −0.131 | 0.371 |
| PMP1 | 3.692 | 1.037 | −0.382 | 0.187 | −0.356 | 0.371 |
| PMP2 | 3.704 | 0.827 | −0.445 | 0.187 | 0.209 | 0.371 |
| PMP3 | 3.834 | 0.852 | −0.727 | 0.187 | 0.941 | 0.371 |
| PMP4 | 3.817 | 0.976 | −0.614 | 0.187 | 0.208 | 0.371 |
| PMP5 | 3.781 | 0.838 | −0.493 | 0.187 | 0.561 | 0.371 |
| PMP6 | 3.811 | 0.975 | −0.49 | 0.187 | 0.127 | 0.371 |
| PMP7 | 3.805 | 1.053 | −0.562 | 0.187 | −0.076 | 0.371 |
| PMP8 | 3.828 | 1.083 | −0.797 | 0.187 | 0.564 | 0.371 |

*3.3. Data Analysis Strategy*

First, the EFA is used to categorize the organizational leadership variables in complex construction projects into multiple dimensions. In this study, the common factor is extracted using the principal component approach, and factor rotation is accomplished using the maximum variance method. When the eigenvalue exceeds 1, a factor is extracted, and vice versa [50].

Second, PLS-SEM is employed to test the measurement and structural model in this study. PLS-SEM is more appropriate for this investigation than covariance-based structural equation modeling (CB-SEM) for two reasons. Initially, PLS-SEM is suited for models that are in the exploratory phase or where the theory needs an additional elaboration [51].

Furthermore, this study includes formative and reflective indicators, and PLS-SEM permits both types of indicators to coexist in a single model [52].

Third, bootstrapping is used to test for mediating effects in this study. Since bootstrapping has a higher statistical validity compared to other methods, it can be applied to medium and small samples and various mediating effect models [53].

## 4. Analysis and Results

### 4.1. Measure Assessment of OL

The results of the EFA of organizational leadership are shown in Table 4. Table 4 reveals that four factors are extracted from organizational leadership variables. The cumulative variance of the characteristic roots of each factor accounts for 49.747% of the total variance, which is near the minimum standard of 50%. The fact that the factor loads are above 0.400 indicates that the four retrieved common factors can accurately depict 14 variables. Factor 1 (including OL2, OL4, and OL10) in Table 4 reflects the characteristics of organizational vision-oriented leadership, which can be named as vision guiding (VG). Factor 2 (including OL5, OL6, OL8, and OL13) reflects the characteristics of organizational system-oriented leadership, which can be named systems thinking (ST). Factor 3 (including OL1, OL3, OL12, and OL14) reflects the characteristics of organizational team-oriented leadership, which can be named as team building (TB). Factor 4 (including OL7, OL9, and OL11) reflects the characteristics of organizational context-oriented leadership, which is named as context interacting (CI). Thereby, the factors of organizational leadership of complex construction projects are identified and shown in Table 5.

Outer weights and $p$ values for the first-order (reflective) dimensions must be reported in accordance with the standards for evaluating formatively models [52,54]. The relevance of organizational leadership (second-order formative) and its dimensions (first-order reflective), as determined by outer weights and P values, are presented in Figure 2 and Table 6. It can be seen from Table 6 that organizational leadership is measured by four dimensions, including vision guiding, systems thinking, team building, and context interacting. Organizational leadership dimensions, i.e., systems thinking ($\beta = 0.349$, $p < 0.01$) and team building ($\beta = 0.349$, $p < 0.01$), show the higher contributions to organizational leadership, followed by context interacting ($\beta = 0.268$, $p < 0.01$) and vision guiding ($\beta = 0.267$, $p < 0.01$).

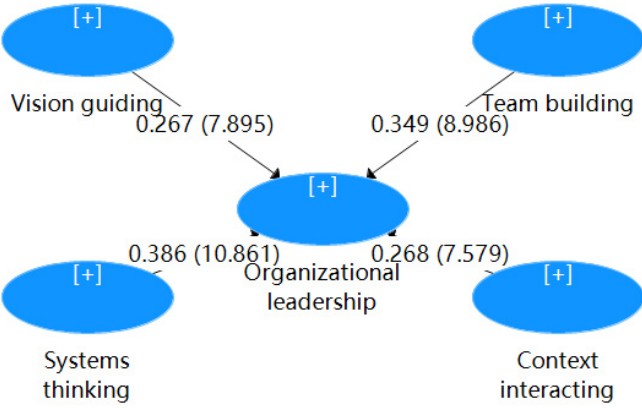

**Figure 2.** Structural model of dimensions of OL.

**Table 4.** Exploratory factor analysis results of organizational leadership.

| Item | Factor | | | | Initial Eigenvalues | | | Extraction Sums of the Squared Loadings | | | Sum of Squares of Rotating Loads | | |
|------|-------|-------|-------|-------|-------|------------------------|-------------|-------|------------------------|-------------|-------|------------------------|-------------|
| | 1 | 2 | 3 | 4 | Total | Percentage of Variance | Cumulative% | Total | Percentage of Variance | Cumulative% | Total | Percentage of Variance | Cumulative% |
| OL1 | 0.185 | 0.432 | 0.469 | −0.002 | 3.286 | 23.475 | 23.475 | 3.286 | 23.475 | 23.475 | 1.961 | 14.004 | 14.004 |
| OL2 | 0.63 | 0.043 | 0.063 | 0.07 | 1.338 | 9.559 | 33.034 | 1.338 | 9.559 | 33.034 | 1.865 | 13.318 | 27.322 |
| OL3 | −0.122 | 0.162 | 0.702 | 0.069 | 1.207 | 8.622 | 41.656 | 1.207 | 8.622 | 41.656 | 1.673 | 11.948 | 39.27 |
| OL4 | 0.551 | 0.303 | 0.131 | 0.182 | 1.133 | 8.09 | 49.747 | 1.133 | 8.09 | 49.747 | 1.467 | 10.476 | 49.747 |
| OL5 | 0.379 | 0.400 | −0.122 | 0.182 | 0.917 | 6.549 | 56.295 | — | — | — | — | — | — |
| OL6 | 0.358 | 0.474 | 0.017 | 0.223 | 0.882 | 6.299 | 62.594 | — | — | — | — | — | — |
| OL7 | −0.011 | 0.154 | 0.05 | 0.757 | 0.854 | 6.1 | 68.694 | — | — | — | — | — | — |
| OL8 | 0.091 | 0.809 | 0.059 | −0.052 | 0.827 | 5.904 | 74.598 | — | — | — | — | — | — |
| OL9 | 0.359 | −0.083 | 0.163 | 0.584 | 0.778 | 5.554 | 80.152 | — | — | — | — | — | — |
| OL10 | 0.76 | −0.005 | 0.067 | 0.066 | 0.676 | 4.83 | 84.982 | — | — | — | — | — | — |
| OL11 | 0.076 | 0.478 | 0.097 | 0.507 | 0.644 | 4.598 | 89.58 | — | — | — | — | — | — |
| OL12 | 0.178 | −0.12 | 0.577 | 0.337 | 0.556 | 3.969 | 93.549 | — | — | — | — | — | — |
| OL13 | −0.131 | 0.488 | 0.347 | 0.095 | 0.471 | 3.362 | 96.912 | — | — | — | — | — | — |
| OL14 | 0.413 | 0.078 | 0.652 | −0.211 | 0.432 | 3.088 | 100 | — | — | — | — | — | — |

**Table 5.** Factors of organizational leadership for complex construction projects.

| Dimensions | Variables |
|---|---|
| Vision guiding (VG) | Development direction (VG1), value orientation (VG2), and resource integration (VG3) |
| Systems thinking (ST) | Dynamic thinking (ST1), system integration thought (ST2), information transfer (ST3), and training system (ST4) |
| Team building (TB) | Cultural identity (TB1), personal qualities (TB2), organization structure (TB3), and incentive system (TB4) |
| Context interacting (CI) | Process of roles (CI1), gentle communication (CI2), and strategy adjustment (CI3) |

**Table 6.** Assessments of formative dimensions of organizational leadership.

| Second-Order (Formative) Construct | First-Order (Reflective) Construct | Path Coefficient | T Statistics | VIF | *p* Values |
|---|---|---|---|---|---|
| Organizational leadership | Vision guiding | 0.267 | 7.895 | 1.495 | 0.000 |
| | Systems thinking | 0.386 | 10.861 | 1.599 | 0.000 |
| | Team building | 0.349 | 8.986 | 1.635 | 0.000 |
| | Context interacting | 0.268 | 7.579 | 1.549 | 0.000 |

### 4.2. Measurement Model of OL, PCB, and PMP

The outer loadings and internal consistency reliability of the factor are used to estimate the reliabilities of the reflective constructs. Table 7 provides the outcomes of the quality criteria needed for the measurement model. It can be seen that maximum items are close to and above the level of 0.7 and 0.8 for all items that met the minimum criteria, i.e., 0.5 for all factors of OL, PCB, and PMP [54,55]. Moreover, Cronbach's $\alpha$ is higher than 0.708 (threshold of 0.7), demonstrating a high level of internal consistency and reliability for each particular indication.

**Table 7.** Assessments of measurement model (reflective constructs).

| Construct | Items | Outer Loadings | Cronbach's $\alpha$ (CA) | Composite Reliability (CR) | Average Variance Extracted (AVE) |
|---|---|---|---|---|---|
| Vision guiding | VG1 | 0.873 | 0.823 | 0.893 | 0.736 |
| | VG2 | 0.821 | | | |
| | VG3 | 0.879 | | | |
| Systems thinking | ST1 | 0.757 | 0.764 | 0.849 | 0.585 |
| | ST2 | 0.777 | | | |
| | ST3 | 0.763 | | | |
| | ST4 | 0.761 | | | |
| Team building | TB1 | 0.788 | 0.756 | 0.844 | 0.576 |
| | TB2 | 0.758 | | | |
| | TB3 | 0.725 | | | |
| | TB4 | 0.766 | | | |
| Context interacting | CI1 | 0.788 | 0.708 | 0.837 | 0.631 |
| | CI2 | 0.758 | | | |
| | CI3 | 0.835 | | | |
| Project citizenship behavior | PCB1 | 0.746 | 0.781 | 0.850 | 0.532 |
| | PCB2 | 0.746 | | | |
| | PCB3 | 0.690 | | | |
| | PCB4 | 0.725 | | | |
| | PCB5 | 0.738 | | | |

**Table 7.** *Cont.*

| Construct | Items | Outer Loadings | Cronbach's α (CA) | Composite Reliability (CR) | Average Variance Extracted (AVE) |
|---|---|---|---|---|---|
| Project management performance | PMP1 | 0.769 | 0.878 | 0.903 | 0.539 |
| | PMP2 | 0.736 | | | |
| | PMP3 | 0.711 | | | |
| | PMP4 | 0.702 | | | |
| | PMP5 | 0.749 | | | |
| | PMP6 | 0.674 | | | |
| | PMP7 | 0.782 | | | |
| | PMP8 | 0.744 | | | |

Convergent validity is assessed with the composite reliability (CR) and average variance extracted (AVE). When the CR value is larger than 0.7 and the AVE value is higher than 0.5, they are regarded as acceptable. The assessment of the constructs in terms of the factor loadings, CR, and AVE are shown in Table 7. The results show that the CR and AVE are higher than 0.837 and 0.532, confirming the constructs' unidimensionality and veracity of the convergent validity.

The square roots of the AVE values should be higher than the correlations between the two separate reflective constructs, according to the discriminant validity criterion put forward by Fornell and Larcker [56]. The values of the model constructs' intercorrelations are in Table 8. The findings support the discriminant validity value of context interacting (0.795), systems thinking (0.765), team building (0.759), vision guiding (0.858), project citizenship behavior (0.730), and the project management performance (0.734) as outweighing the association between every construct.

**Table 8.** Discriminant validity—Fornell–Larcker criterion.

| Construct | Context Interacting | Systems Thinking | Team Building | Vision Guiding | Project Citizenship Behavior | Project Management Performance |
|---|---|---|---|---|---|---|
| Context interacting | 0.795 | | | | | |
| Systems thinking | 0.445 | 0.765 | | | | |
| Team building | 0.392 | 0.447 | 0.759 | | | |
| Vision guiding | 0.462 | 0.518 | 0.474 | 0.858 | | |
| Project citizenship behavior | 0.580 | 0.517 | 0.677 | 0.624 | 0.730 | |
| Project management performance | 0.591 | 0.453 | 0.665 | 0.520 | 0.698 | 0.734 |

Note: Diagonal value represents the square root of AVE, while off diagonal value represents the correlation.

### 4.3. Structural Model of PLS-SEM

The structural model evaluates the statistical significance of all variables' path coefficients [54]. In this study, the PLS-SEM algorithm and bootstrapping procedure are used to test the validity of the structural model. Path coefficients and T values are used to assess the significance level of the structural model [54,57]. The central criterion of the structural model is evaluated by the value of the coefficient of determination $R^2$ [54]. The structural models for OL, PCB, and PMP are shown in Figure 3 and the path coefficients, T values, and significance levels for each factor are shown in Table 9 and Figure 4. Figure 4 shows that the structural model accounted for 67.3% of the variance for PCB and 56.3% of the variance for PMP. The $R^2$ value for this study's model indicates that the parameter estimations have a high level of statistical power. The predictive usefulness of the model is further confirmed using the PLS-SEM blinding process. The derived Stone-value Geisser's for this study ($Q^2 = 0.377/0.472$) satisfies the requirement (i.e., $Q^2 > 0$) for the model's predictive significance [58]. Thus, the PLS-SEM test validates the structural model's fitness.

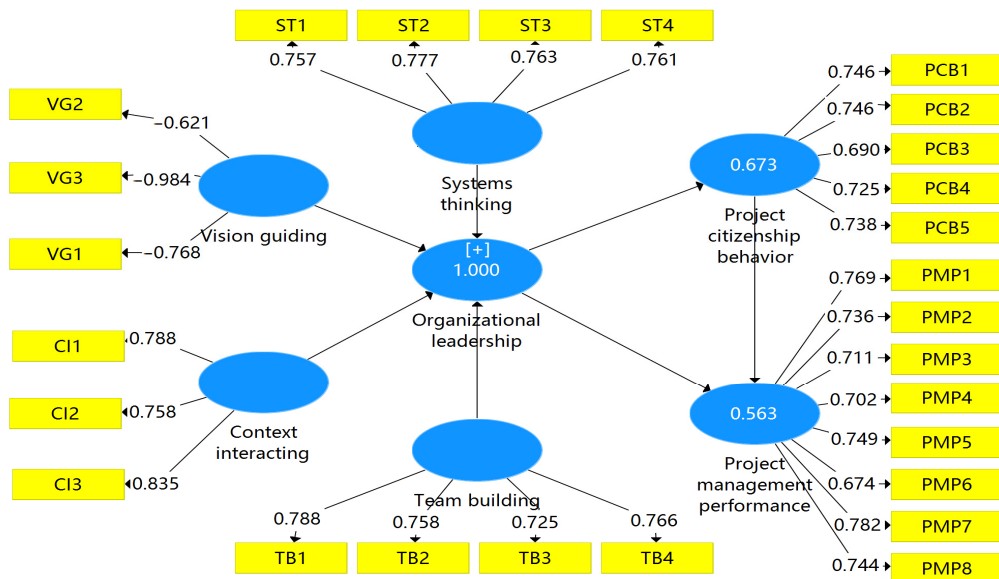

**Figure 3.** Structural model of OL, PCB, and PMP.

**Table 9.** Summary of the structural model.

| Constructs | Path Coefficient | T Statistics | *p* Values | Effect Size (f$^2$) | R$^2$ Value | Q$^2$ Value |
|---|---|---|---|---|---|---|
| OL → PCB | 0.961 | 140.923 | 0.000 | 12.196 | 0.673 | 0.377 |
| OL → PMP | 0.466 | 4.171 | 0.000 | 0.242 | 0.563 | 0.472 |
| PCB → PMP | 0.469 | 4.538 | 0.000 | 0.212 | | |
| OL → PCB → PMP | 0.487 | 4.504 | 0.000 | | | |
| VG → PCB | 0.320 | 3.843 | 0.000 | | | |
| VG → PMP | 0.560 | 5.545 | 0.000 | | | |
| ST → PCB | 0.101 | 1.295 | 0.221 | | | |
| ST → PMP | 0.194 | 1.648 | 0.100 | | | |
| TB → PCB | 0.500 | 5.613 | 0.000 | | | |
| TB → PMP | 0.124 | 1.292 | 0.235 | | | |
| CI → PCB | 0.190 | 2.302 | 0.022 | | | |
| CI → PMP | 0.399 | 2.503 | 0.012 | | | |

Note: Data in the table are in the early stage.

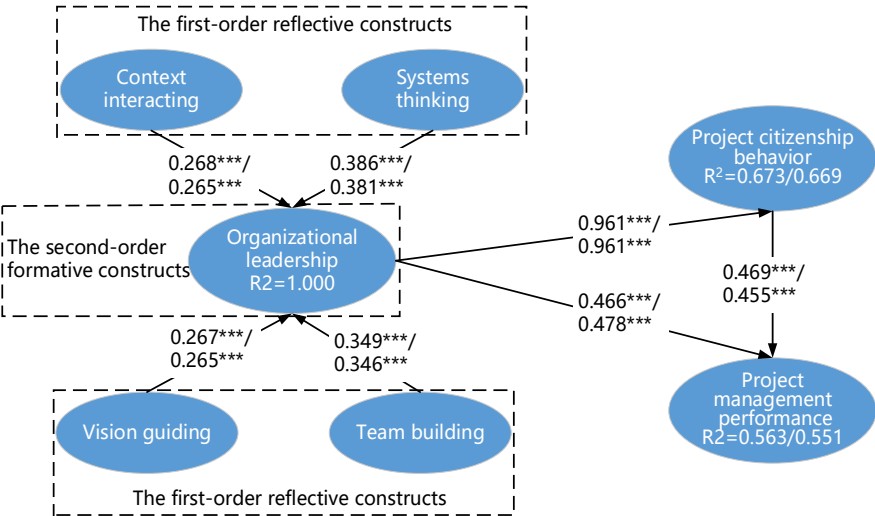

**Figure 4.** The structural model with analysis results. Notes: The early stage/the middle and late stages. *** $p < 0.001$.

*4.4. Hypotheses Testing between OL, PCB, and PMP*

The PLS-SEM bootstrapping approach provides the assessment scores of the structural path model as shown in Table 9. The model shows the effects of OL and PCB on PMP. It can be concluded that stronger organizational leadership has a significant positive effect on the project management performance ($\beta$ = 0.961, T = 140.923, $p$ < 0.01). Thus, H0 is supported. Moreover, that H1 has a stronger organizational leadership has a significant positive effect on project citizenship behavior ($\beta$ = 0.501, T = 3.933, $p$ < 0.01) is supported. Additionally, project citizenship behavior ($\beta$ = 0.469, T = 3.689, $p$ < 0.01) shows significant positive effects on the management performance. Thereby, H2 is supported.

Additionally, the effects of different dimensions of OL on PMP are shown in Table 9. As demonstrated in Table 9, except that the path coefficients of TB → PMP ($\beta$ = 0.124, T = 1.292, $p$ = 0.235), ST → PMP ($\beta$ = 0.194, T = 1.648, $p$ = 0.100), and ST → PCB ($\beta$ = 0.101, T = 1.295, $p$ = 0.221) are not significant, other path coefficients are significantly higher than the 0.05 level. Thus, context interacting and vision guiding have a significant positive impact on the project management performance, while team building and systems thinking have no significant positive impact on the project management performance.

*4.5. Mediating Effects Testing of PCB*

Bootstrapping is adopted in this study to test the mediating effects of project citizenship behavior. A significance level of 0.95 is set and 5,000 subsamples are calculated. The bias-corrected confidence intervals for organizational leadership vary from 0.210 to 0.678 for the early phase and from 0.250 to 0.717 for the middle and late phases. There are considerable indirect effects of organizational leadership on the project management performance since no bias-corrected bootstrapping confidence interval contains 0. Thereby, the mediating role of project citizenship behavior is established, and that H3 project citizenship behavior mediates the effects of organizational leadership on the project management performance is supported. Table 9 provides the path coefficients, T values, and significance level of the factors. As demonstrated in Table 9, the path coefficient of OL → PMP is 0.466 ($p$ < 0.01), and the path coefficient of OL → PCB → PMP is 0.487 ($p$ < 0.01), thus the VAF (variance accounted for) value is 51.1%, indicating that the theoretical model of OL - PCB - PMP is a partial mediation model [55]. Thus, the effect of organizational leadership on the project management performance is mainly realized through the mediating effect of project citizenship behavior.

**5. Discussions**

This study makes significant contributions to the current literature review. First, the existing literature has revealed that it remains poorly understood on the measurement of organizational leadership in complex construction projects. By summarizing the existing research results and Delphi interviews, this study redefines the core connotation of organizational leadership from the perspective of the complexity of complex construction projects. According to the EFA findings, 14 variables may be used to evaluate organizational leadership in complex construction projects, and these variables can be further classified into vision guiding (including OL2, OL4, and OL10), systems thinking (including OL5, OL6, OL8, and OL13), team building (including OL1, OL3, OL12, and OL14), and context interacting (including OL7, OL9, and OL11). The resulting measurement model can be used to further identify the key influencing factors, providing a basis for exploring its influence mechanism on the project management performance. Thus, it is of a great theoretical significance to construct a measurement scale of organizational leadership suitable for complex construction projects.

Second, the PLS-SEM results suggest that stronger organizational leadership has a significant positive effect on the project management performance ($\beta$ = 0.961, T = 140.923, $p$ < 0.01). This is in line with the influence of Mueller et al. [7], Yang et al. [8], and Anantatmula [37] on the relationship between the improvement in leadership and the project management performance. Under the pressure of the epidemic, leaders should increase

communication with participants to build trust to enhance organizational leadership, thus promoting the project management performance. Moreover, stronger organizational leadership has a positive effect on project citizenship ($\beta = 0.501$, T = 3.933, $p < 0.01$), and project citizenship has a positive effect on the project management performance ($\beta = 0.469$, T = 3.689, $p < 0.01$). The conclusion is in line with those of Chan et al. [18] and Wang et al. [38], which showed that stronger leadership can promote project organizational citizenship behavior and the performance of complex projects. Leaders are seen as having a key role when employees fail to know how to respond when faced with timely arrangements. This means that the enhancement of organizational leadership could inspire voluntary behaviors among project members.

Third, this study marks one of the initial efforts to close this important gap by exploring the influence mechanism of organizational leadership on the project management performance from the perspective of complexity, with an eye toward the effect of various dimensions of organizational leadership on the project management performance. The findings help to address improvement dilemmas of the project management performance in a more targeted manner. The results indicate that context interacting ($\beta = 0.399$, T = 2.503, $p < 0.05$) and vision guiding ($\beta = 0.560$, T = 5.545, $p < 0.01$) have a positive effect on the project management performance. This is consistent with the findings of Bennis et al. [13], Kantabutra et al. [14], and Kivipõld et al. [21] that vision orientation and environmental interaction play a positive role in improving performance. In the face of uncertainty, leaders need to make decisions quickly, share information proactively, and strengthen ties with project members. All of these contribute to the formation of project citizenship behavior among project stakeholders, which indirectly affects the project management performance. The findings of this study, however, indicate that systems thinking has no significant impact on project citizenship behavior and the project management performance, which contradicts the results of Palaima [19]. In addition, team building has no significant effect on the project management performance but can improve the project management performance by influencing the project citizenship behavior. These findings suggest that organizational leaders require better communication skills and empathy to help and motivate project members to inspire project citizenship behavior, thus responding quickly to changes

Fourth, the findings highlight the mediating role of project citizenship behavior between organizational leadership and the project management performance. It is noteworthy that the effect of organizational leadership on the project management performance is mainly realized through the mediating effect of project citizenship behavior (VAF = 51%). The results point out the focus of efforts and provide a new perspective for managers in complex construction projects to improve their performance. Specifically, a unanimously accepted organizational culture and the trust gained from a participation in decision-making can motivate the voluntary behaviors of participants to flexibly respond to emergencies during the epidemic, thus motivating project citizenship behaviors to enhance the project management performance in complex construction projects.

## 6. Conclusions and Future Work

This study empirically validates the measurement of organizational leadership in complex construction projects, constructs the correlation model between organizational leadership and the project management performance, and explores the mediating effect of project citizenship behavior. The results show that (i) the organizational leadership of complex construction projects can be effectively measured by vision guiding, systems thinking, team building, and context interacting. (ii) The specific impact of different organizational leadership dimensions on the project management performance are as follows: vision guiding and context interacting have both direct and indirect effects on the project management performance; team building can improve the project management performance by influencing project citizenship behavior, whereas systems thinking has no significant effect on project citizenship behavior and the project management performance

in complex construction projects. (iii) Project citizenship behavior plays a mediating role in the influence of organizational leadership on the project management performance.

The findings of this study provide insights into management practices for complex construction projects. First, the role of vision guiding is critical to the management of complex construction projects. Depicting the organization's vision could inspire management to make changes and innovations to adapt to the complex project environment, which will motivate project members to develop consistent values and generate project citizenship behaviors. Second, regarding context interacting, management could improve their project management performance by positioning the role processes of stakeholders to adjust strategies timely, opening up communication channels between management and project members to improve on management deficiencies. Third, it is an interesting finding that team building improves the project performance by stimulating the project citizenship behaviors of organizational members. The recommendations in this study include active communication and regular group activities to enhance the identification of project participants with the project culture; clarifying and publicizing the incentive system of the project organization; simplifying the organizational structure; and optimizing the workflow, to stimulate project citizenship behavior and thus improve the management performance. Additionally, the study has important theoretical implications. First, this study presents a measured assessment of organizational leadership in complex construction projects that are successfully validated empirically. Existing research lacks the quantitative analysis of organizational leadership from a complexity perspective. The measurement items identified in this study can fill this research gap and provide a framework to explore the influence path on the project management performance. Second, this study explores the mediating role of project citizenship behavior, revealing specific influence paths among organizational leadership, project citizenship behavior, and the project management performance. The findings offer a useful new route for managers of complex construction projects to effectively stimulate their participants' project citizenship behaviors and provide a reference for improving the management performance of such projects. This is a significant addition to the body of literature.

Furthermore, this study has important practical implications. First, the effect of organizational leadership on the project management performance is realized mainly through the mediating effect of project citizenship behavior. Complex construction projects are more difficult to deliver under the impact of COVID-19 and require rapid and comprehensive changes in safety, health, and sanitation needs, relying on leaders to provide clear direction and resources. For example, with senior management actively driving new decisions, leaders need to ensure that employees are involved in decision-making and are committed to putting safety ahead of production. Leadership drives culture, which in turn influences behavior. Through the leadership of project management, project stakeholders can be promoted to produce project citizenship behavior, thus improving the project management performance. Second, from the perspective of the impact path, the project participants' sense of identity in the project culture facilitates project citizenship behavior. During COVID-19, voluntary actions by participants can bridge transient issues and gaps in management that cannot be filled through formal rules and regulations. Thereby, project citizenship behavior is effectively stimulated and the organization's ability to respond to crises is improved. In sum, these conclusions may serve as a reminder to project managers to focus on organizational leadership, foster project citizenship behavior, and ultimately enhance sustainable performance among staff members.

This study has some deficiencies. First, the dynamic effect is not considered in the construction process of complex construction projects during COVID-19. Thereby, a dynamic method should be used to simulate the dynamics of leadership on the project management performance in future research. Second, this study only takes project citizenship behavior into account as a mediating factor in the relationship between organizational leadership and the project management performance. Therefore, subsequent studies can explore the relationship between project citizenship behavior indicators and project management

performance indicators, as well as the mediating or moderating effects of other variables such as the project leadership style and project environment dynamics.

**Author Contributions:** Methodology: L.L.; formal analysis: Y.Y.; data curation: G.W.; writing—original draft preparation: L.L.; review and editing: Y.Y.; investigation: J.Z.; supervision: G.W.; project administration: D.L. All authors have read and agreed to the published version of the manuscript.

**Funding:** This study is supported by the National Natural Science Foundation of China (71901113, 72061025, 71972018, and 72161021), the Natural Science Foundation of Jiangxi Province in China (20212ACB214014), and the Social Science Foundation of Jiangxi Province in China (21GL05).

**Data Availability Statement:** The data presented in this study are available on request from the corresponding author.

**Conflicts of Interest:** There is no conflict of interest.

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
