# Peer review of "Effects of Organizational Leadership on Project Citizenship Behavior and Management Performance in Complex Construction Projects"

_buildings, doi:10.3390/buildings13010259_

Round 1
Reviewer 1 Report
To the authors,
The model constructs are very generic concepts and well researched items hence more studies from the literature should be referenced for the validity of the model. ( Table 1)
Reviewer 2 Report
As a leadership researcher, I will try to provide useful, constructive feedback below.
Overall, the paper lacks of a theoretical foundation particularly in the two domains of organizational leadership and organizational citizenship behavior. There is a whole range of organizational leadership theories in the literature, ranging from vision-based leadership, situational leadership, sustainable leadership. How have you derived at the "organizational leadership" you used in the present study? This needs to be discussed. Otherwise, you cannot demonstrate the contribution of the paper to the broader literature on organizational leadership. Similarly, the concept of organizational citizenship behavior needs to be supported by theoretical concepts/theories in the domain. At it is, it is difficult to justify the contribution of the present study in the subsequent discussion section.
In addition, are you referring to organizational or team leadership? They are not quite the same. Organizational leadership suggests something that lasts longer than team leadership? This needs to be discussed and justified.
Looking closely in the your vision guiding construct, a goal and a vision are not the same. So, is it a vision that you refer to? Or is it indeed a project goal that you refer to? This needs to be discussed and justified in the literature review. You may consult the references below.
Kantabutra, S. (2009), "Toward a behavioral theory of vision in organizational settings", Leadership & Organization Development Journal, Vol. 30 No. 4, pp. 319 337. https://doi.org/10.1108/01437730910961667
Kantabutra S. Toward an Organizational Theory of Sustainability Vision. Sustainability. 2020; 12(3):1125. https://doi.org/10.3390/su12031125.
In terms of constructing the model, OCB is the top level of workplace attitudes that does not take place easily. In theory, employees must have been satisfied with, committed to and identified themselves with their organization first. How do you explain and justify this in your study?
Many missing references throughout the paper. Please check and include the references where needed. Examples are below.
"The impact of project citizenship behavior on performance should not be neglected as well. Some existing studies have used project citizenship behavior as a mediating variable in theoretical models. Few research, meanwhile, has attempted to look into how project citizenship behavior may mediate the relationship between organizational leadership and performance."
"However, few studies have attempted to examine the specific ways in which such effects occur based on a complexity perspective. Complex construction projects are more uncertain, more complex, and more difficult to implement than general projects"
In the methodology section, do you have any control variable? I realize that you collected the data on project size, but did you use it in your subsequent analysis? Naturally, one would assume the project size is a control variable since it can make a difference on the impact. For example, a small project size is easier for the project leader to manage.
In the discussion section, the authors should discuss the findings in relation to the theoretical background and prior studies. How are they similar? How are they different? How do the findings contribute to the theoretical and empirical literature in the domains of organizational leadership and organizational leadership behavior? Significant contributions to the relevant body of knowledge should be explicitly highlighted in this discussion section.
Please also add a section on managerial implications. Practitioners hardly read everything from the beginning of the paper until the end. They look for a section that summarizes the managerial implications of the findings that they can use right away.
Reviewer 3 Report
1, Background information should be added to the beginning of the abstract.
2, It seems complexity is the critical environmental factor. It is suggested to cite more latest articles on complexity. For example,
Song, J., Song, L., Liu, H., Feng, Z., & Müller, R. (2022). Rethinking project governance: Incorporating contextual and practice-based views. International Journal of Project Management, 40(4), 332-346.
Liu, Y., Houwing, E. J., Hertogh, M., Yuan, Z., & Liu, H. (2022). Explorative Learning in Infrastructure Development Megaprojects: The Case of the Hong Kong-Zhuhai-Macao Bridge. Project Management Journal, 53(2), 113-127.
3, Change system thinking into systems thinking.
4, More information should be added about why project management performance and citizenship behavior are worth studying instead of project performance and other behaviors.
5, The recently published PMBOK 7th suggests that anyone working on a project can demonstrate effective leadership traits, styles, and skills to help the project team perform and deliver the required results. The paper seems not in line with this as it emphasizes senior management and leaders. Could you please explain why. What’s the difference between leaders and project managers?
6, The citation style is not consistent. See line 258.
7, The second round of Delphi is not clearly described.
8, PCB is derived from OCB. What about OL?
9, Better not cite publications in other languages.
Round 2
Reviewer 2 Report
Congratulations!
Reviewer 3 Report
The reviewer would like to thank the authors for their responses and related revisions. The reviewer has no further comment on the revised manuscript.